# Small proline-rich proteins (SPRRs) are epidermally produced antimicrobial proteins that defend the cutaneous barrier by direct bacterial membrane disruption

**Chenlu Zhang[1,2]\*, Zehan Hu[3,4], Abdul G Lone[1], Methinee Artami[1], Marshall Edwards[1], Christos C Zouboulis[5], Maggie Stein[1], Tamia A Harris-Tryon[1,3]\***

[1]Department of Dermatology, University of Texas Southwestern Medical Center, Dallas, United States; [2]School of Life Science and Technology, ShanghaiTech University, Shanghai, China; [3]Department of Immunology, University of Texas Southwestern Medical Center, Dallas, United States; [4]State Key Laboratory of Microbial Metabolism, Joint International Research Laboratory of Metabolic & Developmental Sciences, School of Life Sciences and Biotechnology, Shanghai Jiao Tong University, Shanghai, China; [5]Department of Dermatology, Venereology, Allergology and Immunology, Dessau Medical Center, Brandenburg Medical School Theodore Fontane and Faculty of Health Sciences Brandenburg, Dessau, Germany

**\*For correspondence:**
zhangchl1@shanghaitech.edu.cn (CZ);
tamia.harris-tryon@utsouthwestern.edu (TAH-T)

**Competing interest:** The authors declare that no competing interests exist.

**Abstract** Human skin functions as a physical barrier, preventing the entry of foreign pathogens while also accommodating a myriad of commensal microorganisms. A key contributor to the skin landscape is the sebaceous gland. Mice devoid of sebocytes are prone to skin infection, yet our understanding of how sebocytes function in host defense is incomplete. Here, we show that the small proline-rich proteins, SPRR1 and SPRR2 are bactericidal in skin. SPRR1B and SPPR2A were induced in human sebocytes by exposure to the bacterial cell wall component lipopolysaccharide (LPS). Colonization of germ-free mice was insufficient to trigger increased SPRR expression in mouse skin, but LPS injected into mouse skin stimulated increased expression of the mouse SPRR orthologous genes, *Sprr1a* and *Sprr2a*, through activation of MYD88. Both mouse and human SPRR proteins displayed potent bactericidal activity against MRSA (methicillin-resistant *Staphylococcus aureus*), *Pseudomonas aeruginosa*, and skin commensals. Thus, *Sprr1a$^{-/-}$;Sprr2a$^{-/-}$* mice are more susceptible to MRSA and *P. aeruginosa* skin infection. Lastly, mechanistic studies demonstrate that SPRR proteins exert their bactericidal activity through binding and disruption of the bacterial membrane. Taken together, these findings provide insight into the regulation and antimicrobial function of SPRR proteins in skin and how the skin defends the host against systemic infection.

## Editor's evaluation

The reviewers feel that there was significant novelty in the concept that SPRRs directly interface in bacterial host defense. They also felt that there was sufficient rigor, and coupled with the revised narrative, this manuscript is now acceptable for publication.

## Introduction

The skin is the human body's largest organ, with direct contact with the external environment (**Belkaid and Segre, 2014**). As a result, the skin surface continuously encounters a diverse microbial community including bacteria, fungi, viruses, and parasites (**Duerkop and Hooper, 2013**; **Oh et al., 2016**; **Findley et al., 2013**). When host defense is impaired, skin infection results. Thus, skin and soft tissue infections pose a considerable public health threat (**Dryden, 2009**). The majority of infections of the skin are caused by *Staphylococcus aureus* (**Kahn and Goldstein, 2016**). Additionally, *Pseudomonas aeruginosa* infections in burn patients are one of the most common causes of mortality (**Huebinger et al., 2016**). Adding to the challenges of treating infections posed by these pathogens has been the development of antibiotic resistant strains of bacteria such as methicillin-resistant *Staphylococcus aureus* (MRSA) (**Klevens et al., 2007**).

Skin antimicrobial proteins (AMPs) play an essential role in defending the host from the invasion of pathogens (**Zhang and Gallo, 2016**; **Nakatsuji and Gallo, 2012**). Mammalian AMPs are evolutionarily ancient immune effectors that rapidly kill bacteria by targeting bacterial cell wall or cell membrane structures (**Mukherjee et al., 2014**; **Mishra et al., 2018**). Several distinct AMP families, such as β-defensins, cathelicidins, resistin, and S100 proteins, have been identified and characterized in skin (**Gallo and Hooper, 2012**; **Harris et al., 2019**). However, we still have a limited understanding of the arsenal of AMPs expressed by the skin, the regulatory networks that control the expression of AMPs, and how AMPs function to protect mammalian skin surfaces. Even less is known about the contribution that skin appendages make to host defense.

Sebaceous glands (SGs) are specialized epithelial cells that cover the entire skin surface except the palms and soles. SGs excrete a lipid-rich and waxy substance called sebum to the skin surface (**Fischer et al., 2017**; **Zouboulis et al., 2020**; **Zouboulis et al., 2016**). SGs are believed to contribute to the antimicrobial functions of the skin (**Gallo and Hooper, 2012**), yet few current studies have examined the role of SGs in skin host defense. Here, we show the impact of the bacterial cell wall component lipopolysaccharide (LPS) on human sebocytes and demonstrate that specific Toll-like receptor (TLR) ligands stimulate increased expression of members of the small proline-rich protein (SPRR) family. Human SPRR proteins are 6–18 kDa in size and comprise four subclasses (SPRR1, SPRR2, SPRR3, and SPRR4) with a similar structural organization. Among them, two SPRR1 and seven SPRR2 proteins are characterized with a much higher homogeneity, as they contain a similar consensus repeat sequence (**Cabral et al., 2001**). SPRR proteins were originally identified in skin as markers of terminal differentiation that function as substrates of transglutaminase in the crosslinked cornified envelope present at the skin surface (**Cabral et al., 2001**; **Candi et al., 2005**). In this study, we demonstrate that SPRR1 and SPRR2 proteins function as AMPs in the skin. *Sprr1a*−/−;*Sprr2a*−/− mice are more susceptible to MRSA and *P. aeruginosa* skin infection, revealing that SPRR proteins protect against pathogenic bacterial infections of the skin. We also show mechanistically that the bactericidal activity of SPRR proteins is mediated through the binding and disruption of bacterial membranes. Taken together our findings support a novel antimicrobial function of SPRR proteins in skin.

## Results

### SPRR proteins are induced by LPS in human sebocytes

As a first step toward understanding the role of the SG in skin host defense, we performed whole transcriptome RNA-sequencing to compare transcript abundances in human immortalized SG cells (SZ95) treated with LPS to untreated sebocytes (*Figure 1A*). LPS, which coats the surface of Gram-negative bacteria, had broad impacts on gene expression in human sebocytes, including the increased expression of inflammatory cytokines, chemokines, and AMPs (*Figure 1A*). Notably, three members of the small proline-rich family of proteins (SPRR) were markedly upregulated in sebocytes after LPS treatment (*Figure 1A*). To confirm that the expression of SPRR genes were induced by LPS, we used quantitative reverse transcription PCR (qRT-PCR) to analyze the change of the human *SPRR1B*, *SPRR2A*, and *SPRR2D* transcript abundance in SZ95 cells. Consistent with our RNA-seq results, LPS-treated SZ95 cells displayed significant increase in the relative expression of *SPRR* family genes compared to vehicle-treated cells (*Figure 1B*). Additionally, dose–response and time course experiments revealed that 1 µg LPS is optimal for induction of *SPRR* family genes expression (*Figure 1—figure supplement 1*). Consistent with the qRT-PCR results, SPRR proteins expression also increased after LPS treatment

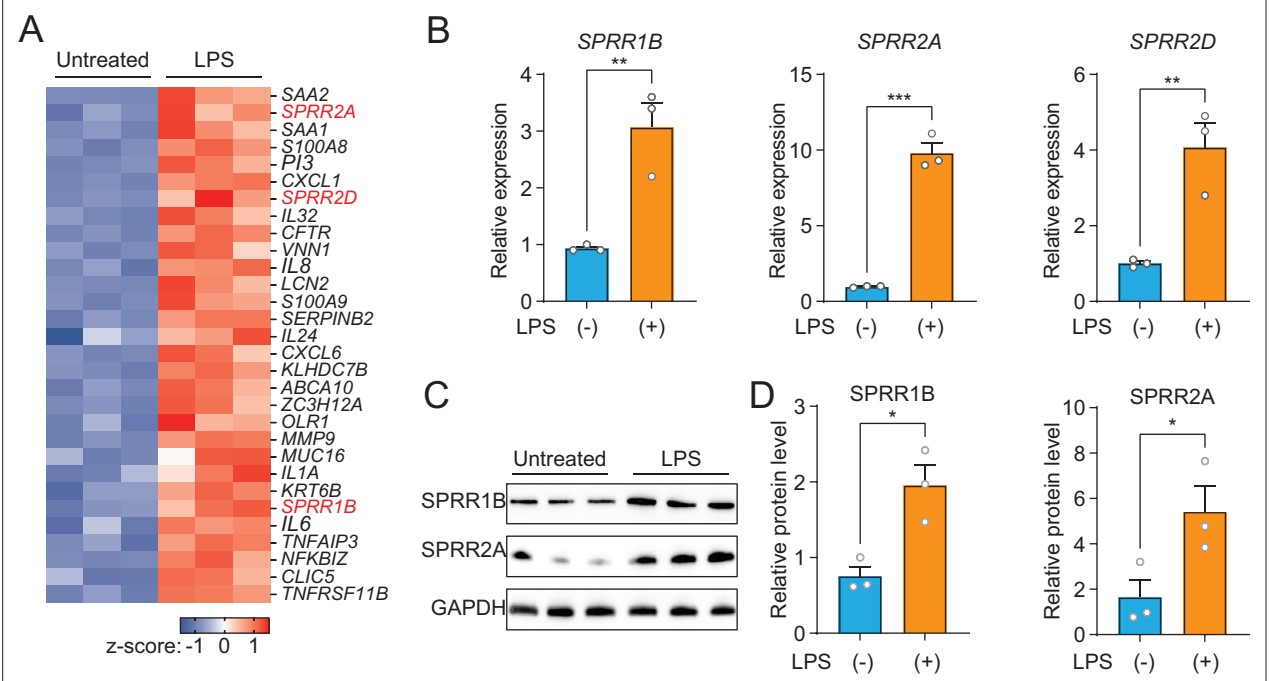

**Figure 1.** The expression of *SPRR* family genes are upregulated by lipopolysaccharide (LPS) in human sebaceous gland cells. (**A**) Heat map of significantly upregulated genes, represented as *Z*-scored RPKM (reads per kilo base per million reads). (**B**) Reverse transcription-quantitative polymerase chain reaction (RT-qPCR) analysis of *SPRR1B*, *SPRR2A*, and *SPRR2D* transcript in the vehicle- and LPS-treated human SZ95 sebocytes. (**C**) Western blot analysis of SPRR1B and SPRR2A was performed on vehicle- or LPS-treated SZ95 cells. GAPDH was used as the loading control. (**D**) Quantification of western blot in (**C**). Means ± standard error of the mean (SEM) are plotted. *p < 0.05, **p < 0.01, and ***p < 0.001 were determined by unpaired *t*-test.

The online version of this article includes the following source data and figure supplement(s) for figure 1:

**Source data 1.** The expression of SPRR proteins increases with lipopolysaccharide.

**Figure supplement 1.** Dose–response and time course analysis of lipopolysaccharide (LPS) treatment on human sebocyte cells.

**Figure supplement 2.** The expression of SPRR family genes are upregulated by Toll-like receptor (TLR)2 and TLR4 agonists in human sebaceous gland cells.

**Figure supplement 3.** Gram-negative bacteria can trigger *Sprr2a* gene expression in human sebaceous gland cells.

in SZ95 cells (*Figure 1C, D*). LPS is a pattern-associated molecular pattern (PAMP) known to trigger gene transcription through the pattern recognition receptor, TLR4 (*Vaishnava et al., 2011*). We therefore tested a panel of PAMPs to see if other foreign stimuli would trigger the expression of the *SPRR* genes in human sebocytes. Interestingly, two ligands of TLR2 (Pam3CSK4 and FSL-1) stimulate the expression of SPRR1B, SPRR2A, and SPRR2D genes in human sebocytes (*Figure 1—figure supplement 2*), indicating that sebocytes respond to these bacterial lipoproteins by expressing SPRR1 and SPRR2 proteins.

Next, we decided to further explore whether heat-inactivated bacteria could induce the expression of SPRR genes in sebocytes. We used qRT-PCR to analyze the change of the human *SPRR2A* transcript abundance in SZ95 cells after treatment with various heat-inactivated bacteria. Interestingly, only the Gram-negative bacteria tested could trigger SPRR2A expression (*Figure 1—figure supplement 3*). Taken together, these data establish that specific bacterial components trigger the expression of SPRR genes in human sebocytes.

## SPRR proteins are upregulated by the injection of LPS in mouse skin

Next, we sought to examine whether commensal skin microbiota colonization could induce the expression of *Sprr* family genes in vivo. *Sprr1b* is not expressed in mouse skin, so we tested the ability of the microbiota to stimulate the expression of the mouse orthologs, *Sprr1a* and *Sprr2a*. In contrast to what we observed in human sebocytes in culture, *Sprr* gene expression was similar between germ-free and conventional mice (*Figure 2A*), indicating that bacterial colonization alone is insufficient to

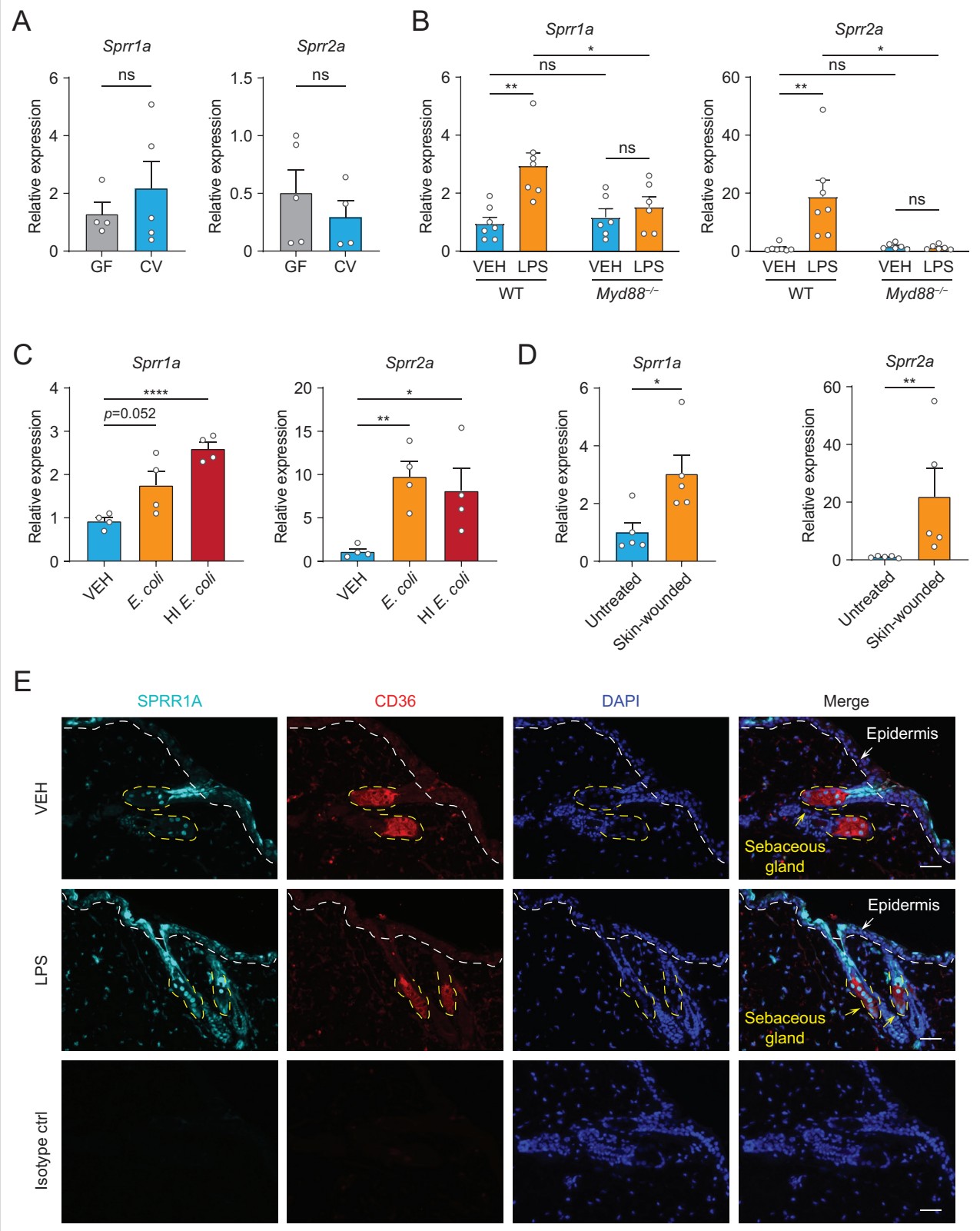

**Figure 2.** The expression of *SPRR* family genes are upregulated by lipopolysaccharide (LPS) in mice. (**A–D**) Quantitative reverse transcription PCR (qRT-PCR) analysis of *Sprr1a* and *Sprr2a* gene expression in mouse dorsal skin tissue. (**A**) Germ-free mice (GF) compared to conventionally raised mice (CV). (**B**) Phosphate-buffered saline (PBS)-treated mouse skin compared to LPS intradermal injection mouse skin in WT or MYD88⁻/⁻ mouse. (**C**) Mouse skin intradermally injected by vehicle (PBS), *E. coli* or heat-inactivated (HI) *E. coli*. (**D**) Untreated mouse skin compared to wounded skin abraded in a

*Figure 2 continued on next page*

*Figure 2 continued*

crosshatch pattern by a 15-blade scalpel. Means ± standard error of the mean (SEM) are plotted. *p < 0.05, **p < 0.01, ****p < 0.0001; ns, not significant by unpaired *t*-test. (**E**) Immunofluorescence staining of SPRR1A expression in mouse skin. CD36 was used as marker of sebocyte cells. Nuclei are stained with 4',6-diamidino-2-phenylindole (DAPI) (blue). Epidermis and sebaceous gland indicated with an arrow. White dashed line separates the epidermis and dermis. Yellow dashed line indicates the outline of SG. Scale bar, 50 µm.

The online version of this article includes the following source data and figure supplement(s) for figure 2:

**Figure supplement 1.** Lipopolysaccharide (LPS) cannot induce the expression of *SPRR* family genes in human keratinocyte cells.

**Figure supplement 2.** Lipopolysaccharide (LPS) cannot trigger the expression of *Sprr* family genes in primary mouse keratinocytes.

**Figure supplement 2—source data 1.** Lipopolysaccharide cannot trigger the expression of SPRR proteins in primary mouse keratinocytes.

trigger increased expression of *Sprr* genes in mouse skin. As the skin surface is coated with waxes, triglycerides and cornified keratinocytes with basal keratinocytes and SGs cells located closer to the dermis, we next chose to test whether deeper exposure to LPS – mimicking wounding and infection conditions — could stimulate *Sprr1a* and *Sprr2a* expression in mouse skin. Indeed, qRT-PCR analysis revealed that injection of LPS stimulates the expression of *Sprr1a* and *Sprr2a* in mouse skin (*Figure 2B*). Canonically, LPS stimulates gene expression through activation of TLR4 and the TLR signaling adaptor MYD88. Thus, LPS injected into the skin of *Myd88*$^{-/-}$ mice did not trigger increased expression of the *SPRR* genes (*Figure 2B*), indicating that LPS induces SPRR protein expression through the TLR-MYD88 signaling pathway. Consistent with these data, both live and heat-inactivated *Escherichia coli* bacteria were able to induce SPRR expression, suggesting that stimulation of *Sprr1a* and *Sprr2a* in mouse skin does not depend on the presence of bacterial metabolites (*Figure 2C*). As has been shown by others (*Vermeij and Backendorf, 2010*), we also found that *Sprr1a* and *Sprr2a* transcript abundance is locally induced by crosshatch skin wounding in mice (*Figure 2D*).

Next, to investigate the expression pattern of SPRR proteins in mouse skin, we performed immunofluorescence using a commercial antibody specific to SPRR1A. SPRR1A is expressed in the SG and nonspecifically at the junction of the SG and hair follicle (*Figure 2E*). This suggests SPRR proteins expressed in the sebocyte may be secreted to the hair canal and are present at the skin surface. Since our immunofluorescence images and prior studies in keratinocytes indicated that SPRR1A and SPRR2A are expressed by keratinocytes (*Fischer et al., 1996*; *Sark et al., 1998*), we sought to study whether LPS could stimulate SPRR expression in keratinocyte cells. We first made use of hTert cells, originally derived from human foreskin keratinocytes and immortalized by expression of human telomerase and mouse Cdk4, by treating the hTert cells with LPS for 24 hr. The qRT-PCR results showed that LPS cannot induce the expression of *SPRR* family genes in human keratinocytes (*Figure 2—figure supplement 1*). To further confirm this result, we isolated primary mouse keratinocytes from neonatal mice, and cultured them under both low and high calcium conditions. Under low calcium conditions, keratinocytes proliferate but require higher calcium conditions to differentiate in culture. Thus, in primary keratinocytes high calcium conditions triggered increased expression of both mouse *Sprr1a* and *Sprr2a* (*Figure 2—figure supplement 2*). However, LPS did not stimulate the increased expression of *Sprr* genes in mouse keratinocytes under either condition (*Figure 2—figure supplement 2*). Taken together, these data show that SPRR proteins are upregulated by LPS in human sebocytes in culture and by LPS injection in mouse skin. Bacterial colonization and LPS treatment of keratinocytes in culture were not sufficient to augment SPRR expression in the skin.

## SPRR proteins are bactericidal against skin pathogens and commensals

The SPRR family of proteins have highly conserved amino acid sequences with numerous cysteine and proline repeats. The antimicrobial properties of proline-rich proteins have not been described in human skin. However, in insects and lower vertebrates, proteins rich in cysteines and prolines have been shown to kill microbes (*Scocchi et al., 2011*; *Stotz et al., 2009*). Based on these findings, we hypothesized that the SPRR proteins might function in cutaneous host defense as AMPs. To test the antimicrobial ability of SPRR proteins, we produced recombinant human SPRR1B, SPRR2A, and mouse SPRR1A protein in the baculovirus insect cell expression system and further purified to homogeneity using size-exclusion chromatography (*Figure 3—figure supplement 1*). We then tested the antimicrobial function of SPRR proteins against skin commensals and pathogens in vitro. When bacteria were exposed to SPRR proteins, we observed a marked dose-dependent reduction in the viability of

*Staphylococcus epidermidis*, MRSA, and *P. aeruginosa* (*Figure 3A*), with a more than 90% decline in bacterial viability after 2-hr exposure to 2.5 μM of SPRR proteins (*Figure 3B*). In contrast, SPRR proteins had no impact on the viability of the prototypical Gram-negative bacteria *E. coli* (*Figure 3—figure supplement 2*). Lastly, to visualize morphological changes of *P. aeruginosa* after exposure to SPRR proteins, we used transmission electron microscopy and observed distinct bacterial cell membrane damage and cytoplasmic leakage (*Figure 3C*).

Since SPRR proteins exhibit similar spectrum of bactericidal activity, we chose to use human SPRR1B and mouse SPRR1A protein to further delineate the mechanism underlying its bactericidal activity. We first used the PI uptake assay to assess the capacity of SPRR proteins to permeabilize bacterial membranes. Human SPRR1B and mouse SPRR1A proteins promoted the dose-dependent uptake of the membrane-impermeant small molecule dye, PI, by *P. aeruginosa* (*Figure 3D*). Next, we examined whether SPRR proteins can directly bind lipid molecules by incubating proteins with lipid strips dotted with different lipids. Both human SPRR1B and mouse SPRR1A bind to lipids bearing negatively charged lipid head groups, but not to neutral lipids (*Figure 3E*). These data are consistent with the idea that most bacterial cell membranes are negatively charged and thus potentially susceptible to SPRR1 binding. Lastly, to determine whether SPRR proteins disrupt bacterial membranes, we incubated human SPRR1B and mouse SPRR1A protein with liposomes loaded with the fluorescent dye carboxyfluorescein (CF, ~10 Å Stokes diameter) and quantified dye release after exposure. Phosphatidylcholine (PC)/phosphatidylserine (PS) liposomes are composed of 85% of the neutral lipid PC and 15% of the negatively charged lipid PS – similar to that of bacterial membranes. Both human SPRR1B and mouse SPRR1A protein-induced rapid dye efflux from negatively charged PC/PS liposomes in a dose-dependent manner (*Figure 3F*). These findings confirm that SPRR proteins exert bactericidal activity through membrane disruption. Taken together, these results suggest that SPRR proteins are bactericidal and compromise the membrane of specific skin bacteria including the pathogen *P. aeruginosa*.

## SPRR proteins limit MRSA and *P. aeruginosa* infection

Given that SPRR1A and SPRR2A proteins exhibited bactericidal activity against a panel of skin pathogens in vitro (*Figure 3*), we predicted that the removal of these proteins might promote MRSA and *P. aeruginosa* skin infection in vivo. To test this hypothesis, we created mice lacking SPRR1A and SPRR2A. We used a CRISPR/Cas9-mediated in vitro fertilization method to delete the entire mouse *Sprr1a* locus of the *Sprr2a*$^{-/-}$ mice (*Hu et al., 2021*) and verified that SPRR1A and SPRR2A are absent in *Sprr1a*$^{-/-}$;*Sprr2a*$^{-/-}$ mice skin at both the transcript and protein level (*Figure 4—figure supplement 1*). *Sprr1a*$^{-/-}$;*Sprr2a*$^{-/-}$ mice were healthy and showed normal skin barrier with no signs of immune infiltration and no significant change in transepidermal water loss (TEWL) (*Figure 4—figure supplement 2*).

To test the *Sprr1a*$^{-/-}$;*Sprr2a*$^{-/-}$ mice for susceptibility to infection, we inoculated the skin of wild-type and *Sprr1a*$^{-/-}$;*Sprr2a*$^{-/-}$ mice with $1 \times 10^6$ CFU of a bioluminescent strain of MRSA by topical application. The luminescence was monitored daily with serial photography and quantified for three consecutive days (*Figure 4A*). Mice devoid of SPRR1A and SPRR2A expression had greater bacterial burdens in the first 3 days of MRSA infection (*Figure 4B, C*). Next, we aimed to assess the susceptibility of *Sprr1a*$^{-/-}$;*Sprr2a*$^{-/-}$ mice to *P. aeruginosa* skin infection. Since damaged or burned skin has increased risk of *P. aeruginosa* infection (*Huebinger et al., 2016*), we first introduced skin wounding in mice by superficial crosshatching method, and then inoculated $1 \times 10^6$ CFU of a bioluminescent strain of *P. aeruginosa* at the wounding site. Intravital imaging and quantification revealed that *Sprr1a*$^{-/-}$;*Sprr2a*$^{-/-}$ mice are more susceptible to *P. aeruginosa* skin infection (*Figure 4D–F*). Collectively, we confirm that SPRR proteins limit MRSA and *P. aeruginosa* infection in mice skin.

## Discussion

Skin is in direct contact with the microbe-filled outer world. Thus, defending the host from invasion by pathogens like *S. aureus* and *P. aeruginosa* is one of the chief functions of the skin (*Kahn and Goldstein, 2016*; *Grice et al., 2008*). In this study, we have identified SPRR1 and SPRR2 proteins as a previously uncharacterized group of antibacterial proteins expressed by keratinocytes and SG cells, which can rapidly kill pathogens through membrane disruption and limit bacterial skin infection.

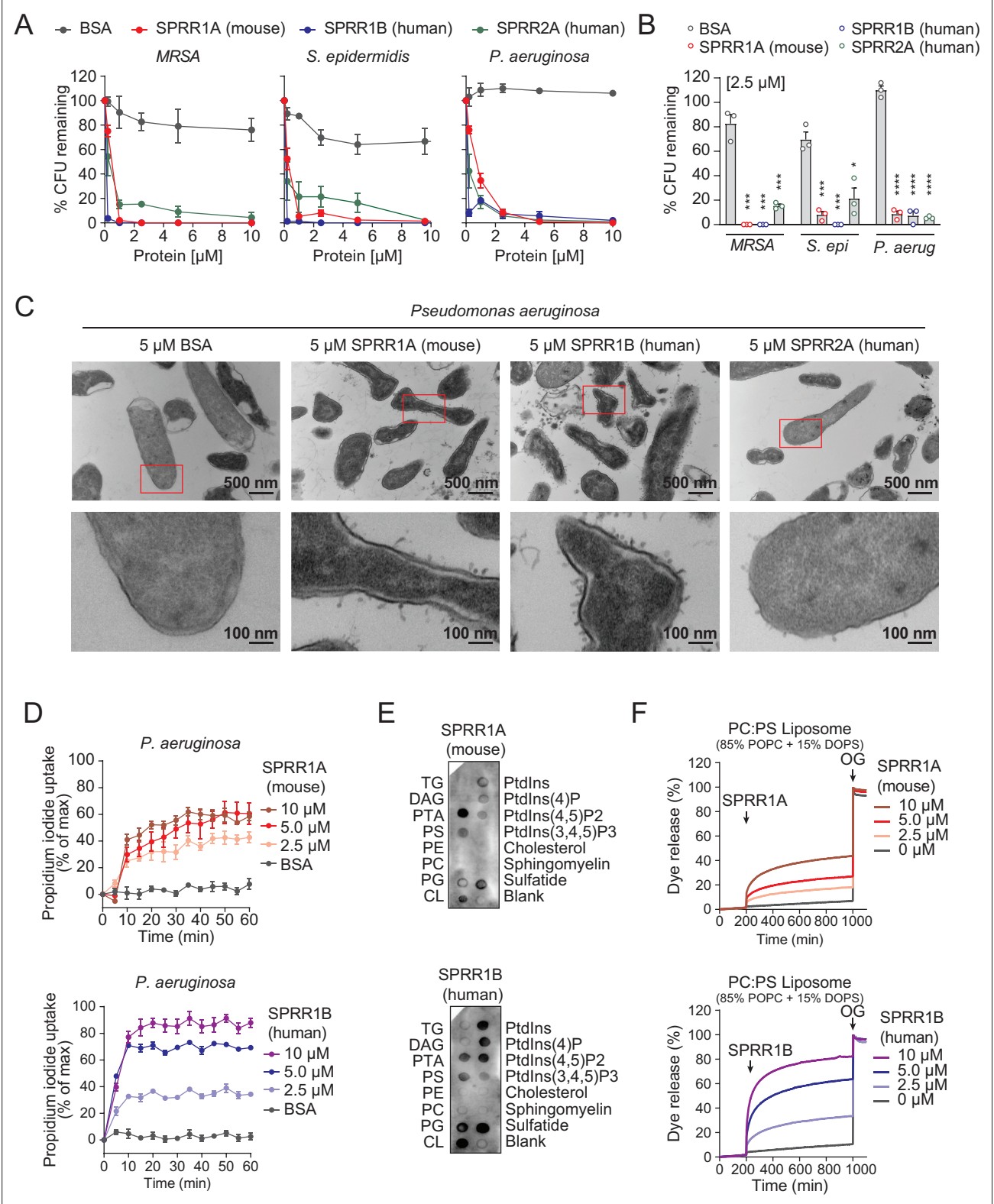

**Figure 3.** SPRR family proteins exert bactericidal activity against various skin commensal and pathogenic bacteria by membrane disruption. (**A**) Increasing concentrations of purified recombinant SPRR proteins were added to mid-logarithmic phase methicillin-resistant *Staphylococcus aureus* (MRSA), *S. epidermidis*, *P. aeruginosa* for 2 hr and surviving bacteria were quantified by dilution plating. (**B**) 2.5 µM of SPRR proteins was added to midlogarithmic phase bacteria for 2 hr and surviving bacteria were quantified by dilution plating. Remaining colony-forming units (CFUs) are expressed as a percentage of untreated bacteria control. (**C**) Transmission electron microscopy of *P. aeruginosa* after incubation with 5 µM purified recombinant

*Figure 3 continued on next page*

*Figure 3 continued*

SPRR proteins. Bovine Serum Albumin (BSA) was used as negative control. Examples of cell surface damage and cytoplasmic leakage are indicated with red rectangular box. Upper scale bars, 500 nm. Lower scale bars, 100 nm. (**D**) Propidium iodide (PI) uptake by *P. aeruginosa* in the presence of increasing concentrations of mouse SPRR1A and human SPRR1B protein. (**E**) Membranes displaying various lipids were incubated with 1 µg/ml SPRR proteins and detected with specific antibody. (**F**) Carboxyfluorescein (CF)-loaded liposomes were treated with increasing concentrations of mouse SPRR1A and human SPRR1B protein, and dye efflux was monitored over time and was expressed as a percentage of total efflux in the presence of the detergent octyl glucoside (OG). All assays were performed in triplicate. Means ± standard error of the mean (SEM) are plotted. *p < 0.05, ***p < 0.001, ****p < 0.0001; ns, not significant by two-tailed *t*-test.

The online version of this article includes the following source data and figure supplement(s) for figure 3:

**Source data 1.** SPRR proteins bind to negatively charged lipids.

**Figure supplement 1.** Recombinant expression and purification of SPRR proteins.

**Figure supplement 1—source data 1.** Coomassie blue staining of fractions in (A) and (C) resolved by SDS-PAGE.

**Figure supplement 2.** SPRR1A protein was resistant to Gram-negative bacteria *Escherichia coli*.

Several earlier studies have shown that SPRR proteins are upregulated in the GI tract, urinary tract, and the airway after exposure to stress and other inflammatory stimuli (*Demetris et al., 2008*; *Hooper et al., 2001*). Further, Hu et al. recently revealed that SPRR2A has antimicrobial actions in the gastrointestinal tract during helminth infection (*Hu et al., 2021*). Our data highlight a previously unappreciated role of SPRR family proteins in skin immune defense, demonstrating that both SPRR1 and SPRR2 are bactericidal proteins induced in sebocytes by the bacterial cell wall component LPS (*Figure 1*). The major role of SGs in mammals is to produce sebum, a mixture of nonpolar lipids and proteins required for normal skin ecology (*Chen et al., 2018*). Sebum secretion can also act as a delivery system for AMPs (*Lovászi et al., 2017*). Though beneficial to the host, AMPs can damage mammalian membranes, so confining expression of AMPs to the nonviable parts of skin through sebum delivery is optimal. Sebum also excretes AMPs to the surface within an acidic milieu, which is often required for AMP activity (*Malik et al., 2016*).

Additionally, our data show that bacterial colonization of germ-free mice does not induce expression of SPRR proteins in vivo (*Figure 2*) and that LPS is not able to stimulate the production of SPRR proteins in mouse keratinocytes (*Figure 2—figure supplements 1–2*). Taken together, our findings suggest that penetration of bacterial stimuli to deeper portions of the skin may be required for MYD88-mediated SPRR protein production in skin. These finding are distinct from what has been observed in the gastrointestinal tract, where colonization of germ-free mice is sufficient to stimulate SPRR2A expression (*Hu et al., 2021*; *Hooper et al., 2001*). As SPRR family proteins also function as crosslinking proteins in terminal differentiation of keratinocytes and formation of the cornified cell envelope (*Candi et al., 2005*), there are likely other regulatory networks that control SPRR protein expression in skin in response to wounding and other stimuli. Additional studies with SG-specific deletion of TLRs and MYD88 will be required to confirm that stimulation of SPRR proteins by LPS requires interaction with TLRs on sebocytes.

In this study, we also reveal that SPRR family proteins have potent bactericidal activity against *Staphylococcal* species and the Gram-negative bacteria *P. aeruginosa* (*Figure 3A–C*), with effective inhibition at low micromolar concentrations. Further, our biochemical data show that mouse and human SPRR1 proteins interact with negatively charged lipid membranes, suggesting broader spectrum antimicrobial capability of these proteins (*Figure 3D–F*). Though SPRR1 was ineffective against *E. coli* (*Figure 3—figure supplement 2*), skin devoid of SPRR1 and SPRR2 was more susceptible to skin infection by MRSA and *P. aeruginosa*, both pathogens that cause skin infections requiring hospitalization (*Figure 4*). Treatments for these infections are currently limited. SPRR proteins might be promising alternatives to traditional antibiotics used to cure MRSA or *P. aeruginosa* infection.

Our findings that SPRR family proteins have antimicrobial function may have implications for other proline-rich proteins expressed in human skin, as the antimicrobial function of proline-rich proteins has been described in other organisms (*Scocchi et al., 2011*; *Stotz et al., 2009*; *Welch et al., 2020*; *Li et al., 2014*). Altogether, these findings expand our current understanding of the molecules involved in cutaneous host defense and provide insight into how the SG contributes to the fight against skin infection.

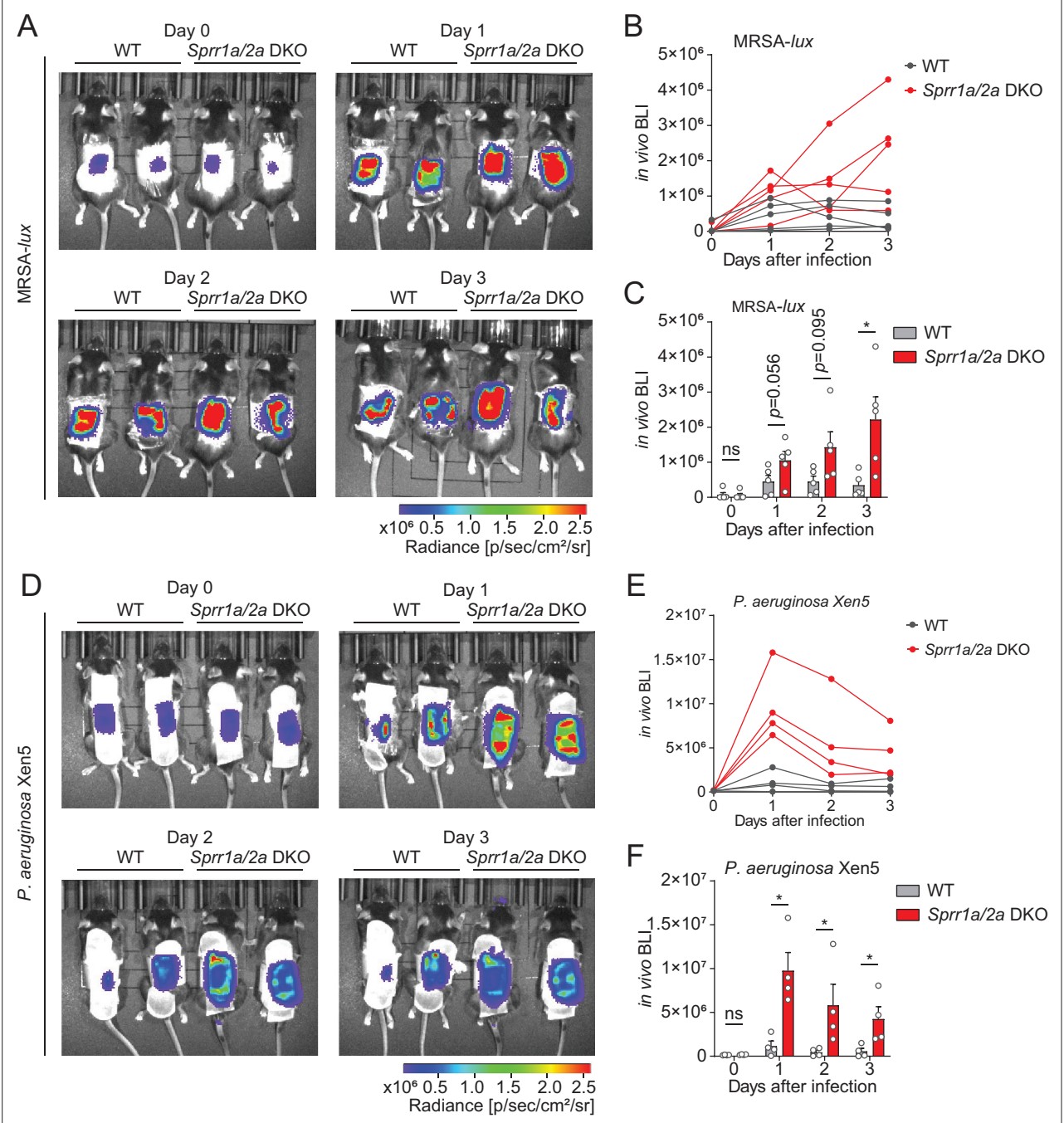

**Figure 4.** SPRR family proteins protect against skin methicillin-resistant *Staphylococcus aureus* (MRSA) and *P. aerugonisa* infection. (**A–C**) WT and *Sprr1a⁻ᐟ⁻;Sprr2a⁻ᐟ⁻* mice were epicutaneously challenged with MRSA (1 × 10⁶ CFU) on the shaved dorsal skin for three consecutive days. (**A**) Representative in vivo bioluminescent imaging (BLI) photographs from days 0 to 3. Quantification of MRSA total flux (photons/s) for each mouse (**B**) or plotted as means ± standard error of the mean (SEM) (**C**). (**D–F**) WT and *Sprr1a⁻ᐟ⁻;Sprr2a⁻ᐟ⁻* mice were superficially abraded in a crosshatch pattern by a 15-blade scalpel to introduce skin wounding. One day post crosshatch wounding, WT and *Sprr1a⁻ᐟ⁻;Sprr2a⁻ᐟ⁻* mice were topically applied with 1 × 10⁶ CFU *P. aerugonisa* on the back skin for 3 days. (**D**) Representative in vivo bioluminescent imaging (BLI) signals from days 0 to 3. Quantification of *P. aerugonisa* total flux (photons/s) for each mouse (**E**) or plotted as means ± SEM (**F**). *p < 0.05, ns, not significant by unpaired *t*-test.

The online version of this article includes the following source data and figure supplement(s) for figure 4:

**Figure supplement 1.** Generation and validation of *Sprr1a⁻ᐟ⁻;Sprr2a⁻ᐟ⁻* mice by CRISPR/Cas9 genomic targeting.

**Figure supplement 1—source data 1.** Western blot analysis of SPRR1A protein in the skin of WT and *Sprr1a⁻ᐟ⁻;Sprr2a⁻ᐟ⁻* mice.

**Figure supplement 2.** *Sprr1a⁻ᐟ⁻;Sprr2a⁻ᐟ⁻* mice does not show signs of inflammation or impaired skin barrier.

## Materials and methods

### Mice

All animal protocols were approved by the Institutional Animal Care and Use Committees of the University of Texas (UT) Southwestern Medical Center. Age- and sex-matched mice 8–12 weeks old were used for all experiments. Conventionally raised C57BL/6 wild type (WT) (RRID: IMSR_JAX:008471), Myd88$^{-/-}$ and Sprr1a$^{-/-}$;Sprr2a$^{-/-}$ on C57BL/6 background were bred and maintained in the specific pathogen-free barrier facility at the UT Southwestern Medical Center on standard chow diet. The generation of Sprr1a$^{-/-}$;Sprr2a$^{-/-}$ mice is described below. Germ-free C57BL/6 mice were bred and maintained in flexible film vinyl isolators in the gnotobiotic mouse facility at UT Southwestern with autoclaved diet (Hooper Lab Diet 6F5KAI, Lab Diet, St. Louis, MO) and autoclaved nanopore water. All mice were housed under a 12-hr light–dark cycle, and mice were randomly assigned to treatment groups.

Sprr1a$^{-/-}$;Sprr2a$^{-/-}$ mice were generated using CRISPR/Cas9-mediated in vitro fertilization with a guide RNA targeting regions upstream and downstream of the Sprr1a locus on the basis of Sprr2a$^{-/-}$ mice (*Figure 4—figure supplement 1*). Sprr2a$^{-/-}$ mice were obtained with the permission of Dr. Lora Hooper (*Hu et al., 2021*). Guide RNAs were injected into fertilized female Sprr2a$^{-/-}$ mice embryos by the UT Southwestern Transgenic Core facility. The resulting litters were screened by genomic sequencing to detect the deletion of Sprr1a and Sprr2a locus. Mice harboring the deleted allele were backcrossed with WT to homozygosity.

### Bacterial strains

Bacteria were grown in species-specific growth media: *E. coli* (ATCC; ATC-PTA-7555) and *Citrobacter rodentium* (ATCC51459) were grown in Luria Broth. *P. aeruginosa* (ATCC27853), bioluminescent *P. aeruginosa* strain Xen05 (Perkin Elmer 119228), and *Enterococcus faecalis* (ATCC29212) were grown in Brain Heart Infusion Broth (BD Biosciences). MRSA (ATCC25923), bioluminescent MRSA strain SAP430 *lux* (*Liu et al., 2017*), *S. epidermidis* (clinical isolate), and *Staphylococcus xylosus* (clinical isolate) were grown in Tryptic Soy Broth.

### Sebocyte cell culture and treatments

SZ95 cells are an immortalized human sebocyte cell line generated from the face of an 87-year-old female and transformed with Simian virus 40 (RRID:CVCL_9803). These cells were previously obtained from Cristos Zouboulis (*Harris et al., 2019*). SZ95 sebocytes were maintained in Sebomed Basal Medium (Fisher Scientific NC9711618) supplemented with 10% fetal bovine serum (GeminiBio 100-106), 5 ng/ml human epidermal growth factor (Thermo Fisher PHG0313), and 1% antibiotic–antimycotic (Gibco 15240062). Cells were cultured in 5% $CO_2$ incubator at 37°C. Cells were stimulated with 1 µg/ml LPS (Sigma L4524). Sixteen hours poststimulation cells were harvested and analyzed as described below. For TLR agonists treatment, reagents in human TLR1-9 Agonist kit (InvivoGen, tlrl-kit1hw) were diluted to working concentration in PBS. 100 ng/ml Pam3CSK4, $1 \times 10^8$ cells/ml HKLM, 500 ng/ml Poly (I:C), 500 ng/ml Flagellin, 100 ng/ml FSL-1, and 1 µg/ml Imiquimod were used for stimulation of SZ95 cells. For heat-inactivated bacteria treatment, bacteria were grown to mid-logarithmic phase, spun down, washed, and resuspended in PBS. Bacteria were heat-inactivated by incubation at 95°C for 20 min. Then $1 \times 10^8$ cells/ml were used for each treatment.

### Immunofluorescence microscopy

Mouse skin samples were fixed in formalin and embedded in paraffin by the UT Southwestern histology core. Samples were deparaffined with xylene followed by rehydration with decreasing concentrations of ethanol. Boiling in 10 mM sodium citrate buffer with 0.2% Tween for 15 min for antigen retrieval. Slides were blocked with 10% fetal bovine serum (FBS), 1% BSA, and 1% Triton X-100 in PBS, and then incubated with primary antibodies against mouse SPRR1A (Thermo Fisher PA5-26062), mouse CD36 (R&D AF2519), rabbit IgG isotype control (Thermo Fisher 02-6102), or goat IgG isotype control (Thermo Fisher 02-6202) using 1:100 dilutions at 4°C overnight. Secondary antibodies Alex Fluor 594 or Alexa Fluor 647 (Thermo Fisher) were diluted 1:350 in blocking buffer and applied to slides for 1 hr at room temperature in the dark. Slides were then washed with PBST (PBS with 0.2% Tween) and mounted with DAPI Fluoromount-G (Southern Biotechnology 0100-20). Images were captured using an Echo Revolve four microscope and cropped in Image J (RRID:SCR_002285).

**Table 1.** Primers for qRT-PCR gene expression analysis.

| Gene | Species | Sequence, 5′→3′ |
| --- | --- | --- |
| *Sprr1a* | *Mus musculus* | Forward: GCCCTGCACTGTACCTCCTC |
| | | Reverse: GTGGCAGGGATCCTTGGTTTT |
| *Sprr2a* | *Mus musculus* | Forward: CCTTGTCCTCCCCAAGTG |
| | | Reverse: AGGGCATGTTGACTGCCAT |
| *Gapdh* | *Mus musculus* | Forward:CACTGCCACCCAGAAGACTGT |
| | | Reverse: GGAAGGCCATGCCAGTGA |
| *SPRR1B* | *Homo sapiens* | Forward: TATTCCTCTCTTCACACCAG |
| | | Reverse: TCCTTGGTTTTGGGGATG |
| *SPRR2A* | *Homo sapiens* | Forward: CCTGAGCACTGATCTGCCTT |
| | | Reverse: GACATGGCTCTGGGCACTTT |
| *SPRR2D* | *Homo sapiens* | Forward: GAGCTAAGAAAAGGAAGTCCTCA |
| | | Reverse: TTATTCAGGGAGTGAACGATAAAT |
| *GAPDH* | *Homo sapiens* | Forward: GGATTTGGTCGTATTGGG |
| | | Reverse: GGAAGATGGTGATGGGATT |

## Quantitative real-time PCR

RNA was extracted from cells or mouse skin using the RNAeasy Plus universal kit (Qiagen 73404). RNA was quantified by absorbance at 260 nm, and its purity was evaluated by the ratios of absorbance at 260/280 nm. 2 µg of RNA was used for cDNA synthesis (Thermo Fisher 4368814, High Capacity cDNA reverse transcription kit). Quantitative real-time PCR was performed using PowerUp SYBR Green Gene Expression Assays (Thermo Fisher A25741) and a QuantStudio 7 Flex Real-Time PCR System (Applied Biosystems). Relative expression values were calculated using the comparative Ct (ΔΔCt) method, and transcript abundances were normalized to GAPDH transcript abundance. The primer sequences are shown in *Table 1*.

## Whole transcriptome sequencing and data analysis

RNA was extracted from SZ95 sebocyte cells using the RNAeasy Plus universal kit (Qiagen 73404). RNA quality was assessed using Agilent 2100 Bioanalyzer. Truseq RNA sample preparation kit v2 (Illumina) was used for the preparation of sequencing libraries. Sequencing was performed on an Illumina HiSeq 2500 (RRID:SCR_016383) for signal end 50 bp length reads. Sequence reads were mapped against the hg19 genome using TopHat. For each gene, read counts were computed using HTSeq-count and analyzed for differential expression using DESeq2.

## Western blot

SZ95 cells were harvested by applying 200 µl of diluted 1× sample buffer (Thermo Fisher 39000) directly to a 6-well plate, scraping down the cell sample to disrupt the membranes, then boiling for 15 min before loading. Equal amounts of protein were loaded onto a 4–20% gradient sodium dodecyl sulfate–polyacrylamide gel electrophoresis (SDS–PAGE) and transferred to a polyvinylidene difluoride (PVDF) membrane. After blocking with 5% milk in TBST, the membranes were incubated with anti-SPRR1B antibody (Thermo Fisher PA5-26062), anti-SPRR2A (Abcam ab125385), or anti-GAPDH (Abcam ab181602) at 4°C overnight. Membranes were then incubated with anti-rabbit secondary antibodies conjugated with HRP (Abcam). Membranes were visualized using a BioRad ChemiDoc Touch system and bands were quantified by Image Lab software (RRID:SCR_014210).

Keratinocyte cell line culture and treatments. hTERT (ATCC CRL-4048) and primary mouse keratinocyte cells were cultured in keratinocyte serum-free medium (KSFM) (Invitrogen, 37010022) supplemented with 0.05 mM CaCl$_2$ (Sigma, C7902), 0.05 µg/ml hydrocortisone (Sigma, H0888), 5 ng/ml epidermal growth factor (EGF) (Invitrogen, 10450-013), 7.5 µg/ml bovine pituitary extract (Invitrogen,

13028-014), 0.5 µg/ml insulin (Sigma, I9278), 100 U/ml penicillin, 100 µg/ml streptomycin, and 25 µg/ml of amphotericin B (Invitrogen, 10450-013). Mouse primary keratinocytes were isolated through dispase digestion. Before digestion, the subcutaneous fat was first removed from the mouse skin. Skin tissue from 3 to 5 days neonatal mice was floated on 1 U/ml dispase (Corning, 354235) in Hank's balanced salt solution (HBSS) (Gibco, 14170) for 16 hr at 4°C with the dermis side down. The next day, the skin was placed in a new dish with the epidermis side down, and the epidermis was peeled and placed into a new dish with HBSS. After being washed, the cells were collected into a new 15 ml tube. The epidermis was cut into small pieces, resuspended in HBSS, gently pipetted up and down several times, and then combined with the cells in the 15 ml tube. The cell solution was filtered with a 70-µm cell strainer, centrifuged at 1000 rpm for 4 min, and resuspended in complete KSFM medium. The cells were gently washed once and seeded in culture dishes with complete KSFM medium. The culture dishes were precoated with collagen (Advanced Biomatrix, 5005-B). Keratinocytes from each mouse were seeded in one 10 cm dish, and fresh complete KSFM medium was supplied after 24 hr. After 3–4 days, the primary keratinocytes reached approximately 80% confluence under normal culture conditions in a 5% $CO_2$, 37°C incubator. For keratinocyte differentiation, cells were treated with 1 mM $CaCl_2$ for 2 days. For treatment of keratinocyte cells, 1 µg/ml LPS (Sigma L4524) were supplemented in the KSFM complete medium. Sixteen hours poststimulation, cells were harvested for qRT-PCR or western blot analysis.

## Intradermal injection of mice

The mouse dorsal hair was removed by shaving (Andis ProClip), followed by depilatory cream (Nair) 1 day before injection. LPS (Sigma L4524) was dissolved in PBS and further diluted to a concentration of 1 mg/ml. For intradermal injection of LPS, each mouse was injected with 50 µl of LPS solution to the back skin. For intradermal injection of bacteria, various bacteria were grown in species-specific media to mid-logarithmic phase, spun down, washed and resuspended in PBS to a concentration of $1 \times 10^8$ cells/ml. Bacteria were heat-inactivated by incubation at 95°C for 20 min. 100 µl of live or heat-inactivated bacteria was injected to the mouse back skin. The site of injection was circled with permanent marker. After 8 hr, mice were sacrificed and the injection site skin was analyzed.

## Protein expression and purification

*SPRR* genes containing a C-terminal 6xHis tag were cloned into a pFastBac1 vector and heterologously expressed in Sf9 cells (Thermo Fisher). One liter of cells ($2.5 \times 10^6$ cells/ml) was infected with 10 ml baculovirus at 28°C. Cells were cultured in suspension and harvested 48 hr after infection. Harvested cells were resuspended in the buffer containing 25 mM Tris–HCl, pH 8.0, 150 mM NaCl, and 1 mM phenylmethylsulfonyl fluoride, and lysed by sonication. The mixture was pelleted by centrifugation at $10,000 \times g$ for 30 min and the supernatant was loaded onto a $Ni^{2+}$ metal affinity column (Qiagen). The column was washed with buffer containing 30 mM imidazole and the protein was eluted in buffer containing 300 mM imidazole. The eluate was concentrated in a 3 K cutoff Amicon Ultra centrifugal device (Millipore) and further purified by size-exclusion chromatography on a Superdex 75 10/300 GL column (GE Healthcare Life Sciences) in standard assay buffer (10 mM 2-(N-morpholino)ethanesulfonic acid (MES), pH 6.0, and 25 mM NaCl).

## Bacterial killing assays

Bacteria were grown in species-specific growth media as described above. 10 ml bacterial cultures were grown to mid-logarithmic phase and then pelleted and washed twice in assay buffer (10 mM MES, pH 6.0, and 25 mM NaCl). Approximately $5 \times 10^6$ cells/ml bacteria were then incubated at 37°C for 2 hr in assay buffer with varying concentrations of recombinant SPRR protein or BSA (Gemini 700-106P). Surviving colony-forming units (CFUs) were quantified by dilution plating onto agar plates and calculated as a percentage of the remaining colonies in the assay buffer only control sample.

## Lipid strip assay

Membrane lipid strips (Echelon, P-6002) were used following the manufacturer's protocol. Briefly, the lipid strips were blocked with blocking buffer (10 mM MES, pH 6.0, 25 mM NaCl, 2% BSA, and 0.05% Tween-20) for 1 hr at room temperature. Purified recombinant SPRR proteins were diluted to 1 µg/ml in blocking buffer and incubated with the lipid strip overnight at 4°C. After three washes with washing

buffer (10 mM MES, pH 6.0, 25 mM NaCl, and 0.05% Tween-20), the lipid strip was sequentially incubated with anti-SPRR1B antibody and HRP-conjugated secondary antibody. Dots were detected with ECL reagent (BioRad, 1705060) using a BioRad ChemiDoc system.

## Electron microscopy

For electron microscopy of bacteria, 10 ml *P. aeruginosa* cultures were grown to mid-logarithmic phase and then pelleted and washed in 10 ml standard assay buffer (10 mM MES, pH 6.0, and 25 mM NaCl). Bacteria were then resuspended in 1 ml standard assay buffer. Purified recombinant SPRR proteins was added at a final concentration of 5 µM to 300 µl of resuspended bacteria and incubated at 37°C for 2 hr. Bacteria were centrifuged for 10 min at 16,000 × *g*, resuspended in crosslinking reagent (4% paraformaldehyde and 4% glutaraldehyde in 0.1 M sodium phosphate buffer, pH 7.4), and incubated overnight at 4°C. After three washes, bacterial pellets were embedded in 3% agarose, sliced into small blocks (1 mm$^3$), and fixed with 1% osmium tetroxide and 0.8% potassium ferricyanide in 0.1 M sodium phosphate buffer for 1.5 hr at room temperature. Cells were rinsed with water then stained with 1% uranyl acetate in water for 1 hr. Cells were dehydrated with increasing concentrations of ethanol, transitioned into propylene oxide, infiltrated with Embed-812 resin, and polymerized overnight in a 60°C oven. Blocks were sectioned with a diamond knife (Diatome) on Leica Ultracut seven ultramicrotome (Leica Microsystems), collected onto copper grids, and poststained with 2% aqueous uranyl acetate and lead citrate. Images were captured on a Tecnai G2 spirit transmission electron microscope (Thermo Fisher) equipped with a LaB6 source using a voltage of 120 kV.

## Liposome disruption assay

Unilamellar liposomes were prepared using lipids from Avanti. 85% 1-palmitoyl-2-oleoyl-*sn*-glycero -3-phosphocholine or POPC (AVANTI PC #850457C) and 15% 1,2-dioleoyl-*sn*-glycero-3-phospho-L -serine or DOPS (AVANTI PS #840035C) dissolved in chloroform were mixed in specific molar ratios in glass tubes and then dried under a stream of $N_2$, followed by drying under vacuum overnight to ensure complete removal of organic solvents. Dried lipids were then combined with 5 (6)-CF (Sigma #21877) and vortexed for 5 min. Lipids were transferred to 2 ml cryotubes and subjected to five freeze–thaw cycles in liquid $N_2$ and stored at −80°C freezer. Lipids were then thawed and passed through a 100 nm pore membrane using a mini-extruder kit (Avanti Polar Lipids #610000) and purified on a PD-10 column to remove excess dye. Prepared liposomes were diluted in standard assay buffer (10 mM MES, pH 6.0, and 25 mM NaCl) to a working concentration of 100 µM. QuantaMaster 300 fluorometer (Photon Technology International) was used to monitor fluorescence. SPRR protein was added to the system at increasing concentrations. At the end time point, 1% (vol/vol) *n*-octylglucoside detergent (OG, Anatrace #O311) was added to completely disrupt the liposomes. Fluorescence was measured over time in seconds and as a percentage of total CF dye released by the detergent OG.

## Dye uptake assay

Bacterial cultures were grown to mid-logarithmic phase and then pelleted and washed in standard assay buffer (10 mM MES, pH 6.0, and 25 mM NaCl). Bacteria were then diluted to approximately $5 × 10^8$ cells/ml in standard assay buffer containing 5.5 µg/ml propidium iodide (PI) (Thermo Fisher, P3566). Then bacterial samples (90 µl each well) were added to black 96-well Costar plates (Fisher, 07-200-567) and placed into a Spectramax plate reader (Molecular Devices) that was pre-equilibrated to 37°C for 10 min. After an initial reading, 10 µl of recombinant purified SPRR proteins at varying concentrations or BSA were added and fluorescence outputs (excitation, 535 nm; emission, 617 nm) were measured every 5 min for 1 hr. PI uptake activity was measured against the maximum fluorescence output from the positive control added with 0.05% SDS.

## Skin infections

The dorsal hair was removed from C57BL6 *Sprr1a*$^{−/−}$;*Sprr2a*$^{−/−}$ or WT male mice by shaving (Andis ProClip), followed by depilatory cream (Nair). For MRSA infection, 1 day after shaving, bioluminescent MRSA strain SAP430-*lux* were grown to mid-logarithmic phase in Tryptic Soy Broth and then pelleted and washed twice in PBS. Approximately $1 × 10^6$ CFU bacteria in 100 µl PBS were placed on rectangular gauze. For *P. aeruginosa* infection, after 24 hr, the dorsal skin was superficially abraded in a crosshatch pattern by a 15-blade scalpel (Fine Scientific Tools 10115-10) to introduce skin wounding.

One day postwounding, the bioluminescent *P. aeruginosa* strain Xen05 (Perkin Elmer 119228) were grown in Brain Heart Infusion Broth (BD Biosciences) and around $1 \times 10^6$ CFU bacteria in 100 µl PBS were placed on a gauze rectangle. The gauze was applied to the dorsal skin of *Sprr1a*$^{-/-}$;*Sprr2a*$^{-/-}$ or WT mice and held in place with two tegaderm (3 M 9505 W) and a Band-Aid Sheer Strips (BAND-AID) for 3 days. The luminescent was monitored daily with serial photography and quantification using a IVIS Lumina3 imager machine.

## TEWL measurement

TEWL of mice dorsal skin was measured using Vapometer (Delfin Technologies) according to the manufacturer's instructions.

## Statistical analysis

Statistical details of experiments can be found in the figure legends, including how significance was defined and the statistical methods used. Data represent mean ± standard error of the mean. Formal randomization techniques were not used; however, mice were allocated to experiments randomly and samples were processed in an arbitrary order. Mouse skin samples that were determined to be in the anagen hair cycle were excluded. All statistical analyses were performed with GraphPad Prism software Version 7.0 (RRID:SCR_002798). To assess the statistical significance of the difference between two treatments, we used two-tailed Student's *t*-tests. To assess the statistical significance of differences between more than two treatments, we used one-way analysis of variance.

## Acknowledgements

We would like to thank Dr. Lora Hooper for helpful discussion and sharing *Sprr2a*$^{-/-}$ mice strain. We thank Dr. Richard Wang for sharing the hTert immortalized human keratinocyte cell line. We thank Dr. Zheng Kuang for his help with analysis of RNA-sequencing data. This work was supported by a Harold Amos Award through the Robert Wood Johnson Foundation, National Institutes of Health K08 award, a UT Southwestern Disease Oriented Clinical Scholars Program award, and a Burroughs Wellcome Fund Career Award for Medical Scientists.

## Additional information

### Funding

| Funder | Grant reference number | Author |
| --- | --- | --- |
| National Institutes of Health | AR076459-01 | Tamia A Harris-Tryon |
| Robert Wood Johnson Foundation | Amos | Tamia A Harris-Tryon |
| Burroughs Wellcome Fund | CAMS | Tamia A Harris-Tryon |

The funders had no role in study design, data collection, and interpretation, or the decision to submit the work for publication.

### Author contributions

Chenlu Zhang, Conceptualization, Formal analysis, Investigation, Methodology, Supervision, Validation, Visualization, Writing – original draft; Zehan Hu, Formal analysis, Investigation, Methodology, Resources, Writing - review and editing; Abdul G Lone, Formal analysis, Investigation, Methodology, Visualization; Methinee Artami, Formal analysis, Investigation, Supervision; Marshall Edwards, Investigation, Supervision; Christos C Zouboulis, Resources, Writing - review and editing; Maggie Stein, Investigation; Tamia A Harris-Tryon, Conceptualization, Formal analysis, Funding acquisition, Methodology, Resources, Supervision, Validation, Visualization, Writing – original draft, Writing - review and editing

## Author ORCIDs
Chenlu Zhang (ID) http://orcid.org/0000-0001-9462-9237
Marshall Edwards (ID) http://orcid.org/0000-0002-8560-1854
Christos C Zouboulis (ID) http://orcid.org/0000-0003-1646-2608
Tamia A Harris-Tryon (ID) http://orcid.org/0000-0002-4170-7083

## Ethics

This study was performed in strict accordance with the recommendations in the Guide for the Care and Use of Laboratory Animals of the National Institutes of Health. All of the animals were handled according to approved institutional animal care and use committee (IACUC) protocols of the University of Texas Southwestern, protocol number 2015-101064. All surgery was performed under isoflurane anesthesia, and every effort was made to minimize suffering.

## Decision letter and Author response
Decision letter https://doi.org/10.7554/eLife.76729.sa1
Author response https://doi.org/10.7554/eLife.76729.sa2

## Additional files

### Supplementary files
• Transparent reporting form

### Data availability
RNA-seq data (Figures 1B,1C) has been submitted to the Gene Expression Omnibus with an accession number: GSE182756.

The following dataset was generated:

| Author(s) | Year | Dataset title | Dataset URL | Database and Identifier |
|---|---|---|---|---|
| Zhang C, Harris-Tryon T | 2022 | Small proline-rich proteins 1 and 2 function as antimicrobial proteins in the skin | https://www.ncbi.nlm.nih.gov/geo/query/acc.cgi?acc=GSE182756 | NCBI Gene Expression Omnibus, GSE182756 |

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
