## [Editor Report]

The reviewers feel that there was significant novelty in the concept that SPRRs directly interface in bacterial host defense. They also felt that there was sufficient rigor, and coupled with the revised narrative, this manuscript is now acceptable for publication.

---

## [Decision Letter]

[Editors' note: this paper was reviewed by Review Commons.]

---

## [Author Response]

Reviewer #1 (Evidence, reproducibility and clarity (Required)):1. Summary:The authors present a solid study on the antimicrobial effects/potential of SRPPs. In general, the study seems very well prepared and is rather dense in its repertoire of techniques and assays. However, there are some minor elements, which I think should be addressed prior to publication.

We thank the reviewer for their review of our work and their positive comments regarding the quality of the data presented and the “repertoire of techniques” used to justify our conclusions. We have addressed the “minor elements” that were suggested in our revised manuscript and these are detailed below, point by point.

2. Major elements:Line 135: SPRR proteins are upregulated by the injection of LPS in mouse skin. This heading is confusing and misleading given that the paragraph is rounded off with the concluding remark that "SPRR proteins are unregulated by LPS in human sebocytes in culture and by LPS dermal injection in mouse skin." Is this not contradictory?The overall impression of the study is that not all the pieces of the puzzle fits, and the authors do also indicate this. However, certain elements could maybe be clarified and the authors stand could be justified more clearly with references to previous work. For example; line 255 "Taken together, our findings suggest that deeper penetration of bacterial stimuli may be required for MYD88 mediated SPRR protein production in skin." It is not clear to me how the authors ended up with this conclusion. SPRR can be expressed in keratinocytes, the sebaceous gland and the fibroblasts (likely also other places). Keratinocytes lying on top and fibroblasts forming the "deeper tissue" which the authors refers to, or at least that is how I would interpret this. So the conclusion in other words would be that SPRR expression from fibroblasts is needed to gain proper protection. But I'm not sure that is really what the authors want to convey.

We thank the reviewer for these thoughtful comments. We have revised this section of the manuscript to describe our findings more clearly. Please see Lines 143-186 of the revised manuscript. Our larger point was that bacterial colonization alone is insufficient to trigger augmented SPRR expression, likely because the skin surface is somewhat inert and bacterial products need to penetrate the cornified surface in order for sebocytes or other proliferating cells to be exposed to bacterial signals. We have revised this section of the manuscript to summarize this point more clearly.

3. In the discussion it is stated that "biochemical data show that mouse and human SPRR1 proteins interact with negatively charged lipid membranes, suggesting broad spectrum antimicrobial capability of these proteins (Figure 3D-F)." I would disagree with this. *E. coli* is one of the easiest gram-negative bacteria one can kill with AMPs, and the authors have already stated that *E. coli* is not affected by SPRR1A. So I think the statement needs to be toned down, and or at least the lack of effect against *E. coli* should be stated again. Furthermore, it is super fascinating that SPRR1A is resistant to *E. coli*, and rather cool to learn that you uses LPS from *E. coli* to stimulate SPRR. Which prompts the question, would LPS from a different source, be more potent in stimulating SPRR, thus changing the entire story. This last comment is more for further work and should not be addressed in the revised version of course.

We thank the reviewer for these thoughtful comments. We have revised the manuscript to again state that *E. coli* is resistant to SPRR killing – Line 287 of the revised manuscript.

We agree that how LPS from different sources stimulates SPRR expression is a question of interest. As a first step, we did investigate the ability of an array of heat killed bacteria to stimulate SPRR expression in cells. These data showed that several heat-killed Gram-negative bacteria were able to stimulate SPRR2A expression in cells. However, the Gram-positive bacteria tested did not have this ability (Supplemental Figure S3). These in vitro data on single strains of Gram-negative bacteria did show a greater impact of heat-killed *E. coli* compared to strains of *P. aeruginosa* and *C. rodentium.*

4. Elements from the SPRR classification line 274 was could ideally have been added in the introduction, which would require less googling for readers with less background knowledge on SPRR.

We thank the reviewer for this excellent suggestion and have moved these sections to the introduction – Lines 96-99.

5. The section on proline-rich proteins as antimicrobials (line 277 to 281) stand in its current for as something unique and novel, however, the literature is rich in papers describing the antimicrobial effect of proline rich AMPs. Thus, justifying rephrasing of the text in the manuscript and also maybe addition of a reference of two.

We thank the reviewer for these suggestions and have revised this section and added a recent review on proline-rich proteins as antimicrobials. Line 298-299

6. The figures are overall of exceptional quality. However, the figure legend for Figure 1 is somewhat misleading. (B) is not a volcano plot. The heat map is (A and not C), "SPRR1B, SPRR2A and SPRR2D are highlighted in red" this info is redundant. The western blot is missing… etc.

We thank the reviewer for their comment that our “figures are overall of exceptional quality”. We also thank the reviewer for catching the error in the figure legend for Figure 1 and for their suggested edits. We have made the requested changes to the figure legend. Line 569-580.

7. Minor elementsIt's a minor comment. However, it is speculative to argue that 16h is the optimal for induction of SPRR. Expression of SPRR will most likely follow a bell-shaped curve. For SPRR1B at 1ugLPS the top of the curve is reached after 16h. However, this is clearly concentration dependent. Figure S1a, also illustrates that at lower LPS concentration, expression is less pronounced, thus it will take longer to reach max expression.

We thank the reviewer for their comments and have modified the text. Line 124-125

8. Figures are referenced with both capital F and small f (unify)

We thank the reviewer for their suggestion. In the original version, lower case “f” was for supplemental figures. We have revised the manuscript to clarify the call outs for supplemental figures.

9. Always use italic on Latin phrases.

We thank the reviewer for their thorough review. We have italicized Latin phrases in the revised manuscript.

10. Line 168, write out "would not"

We thank the reviewer for this suggestion and have reworded this sentence. Line 178-181

11. Candi et al. should be reference properly Line 260

We thank the reviewer for catching this and have updated the reference – Line 277-278

Reviewer #1 (Significance (Required)):The role of SPRRs in human health is not well described. It's a beautiful study, comprehensive, but at the same time clearly painting the picture that more work is needed.It is not surprising that SPRR harbor antimicrobial potential, one could easily envision this potential by comparing the primary sequence of these molecules with other well described proline rich AMPs. The authors have also previously published on this antimicrobial effect, so the news value of that is not very high, however, to figure out how they work, how they are stimulated etc to pinpoint their relevance in human health is obviously interesting and has new value.

We thank the reviewer for their thoughtful comments and reflection on the “new value”, “relevance”, and “[obvious] interest” of our work. We thank the reviewer as well for their comment that ours is a “beautiful” and “comprehensive” study.

Reviewer #2 (Evidence, reproducibility and clarity (Required)):1. A most elegant study that shows how bacterial compounds can stimulate sebocytes to secrete a group of bactericidal proteins that than can kill different Gram-positive and Gram-negative bacteria. The most impressive part is the finding that colonization by itself does not induce secretion of these proteins. Authors describe that the depth of penetration is determining the secretion yes or no. This is in line with their findings that all kind of TLR-stimulants do induce expression and secretion, that would include compounds of commensal bacteria. This correlation with depth of penetration means that the way the system discriminates between harmless colonizers and pathogens is by "measuring" the depth of penetration into the skin. In my view this aspect can be addressed in more detail in abstract and discussion.

We thank the reviewer for their comments on the elegance of our study and that “The most impressive part” is the finding that colonization by itself does not induce these proteins. We have revised the abstract and the manuscript to highlight these data and make these findings clearer to the reader. Lines 53-54 and Lines 143-186.

2. Authors should make an estimate, based on measurements, of how much protein is released in the skin and relate this to the effective concentration found in the bacterial killing experiments. The question is: does this attribute to in vivo killing, as is indicated in the KO-mouse or is it one of very many things that attribute to killing?

We thank the reviewer for their comments. Quantifying the amount of a given skin protein is challenged by the difficulty of solubilizing epidermal proteins. Further, dissecting specific sub compartments of the skin to quantify the concentration of proteins within secretory ducts and at the skin surface—as has been done in paneth cells of the gastrointestinal tract (Ayabe et al., 2000)— remains to be achieved. In our current study, we were able to partially solubilize skin tissue for western blotting and show in Supplemental Figure S8C that the SPRR1 antibody binds to a large band of protein in mouse skin. Immunofluorescence in Figure 2E also shows the brightest signal for SPRR1 at the skin surface.

Given the challenges of quantifying skin proteins, we can estimate the expression by comparing the RNA expression of SPRR proteins to other known antimicrobial proteins. Author response table 1 is a chart of FPKM values (Fragments Per Kilobase of transcript per Million mapped reads) of mouse *SPRR1a* and *SPRR2a* from our published RNA-sequencing data set (Harris et al., 2019). In the skin of mice housed in an SPF facility at UT Southwestern, SPRR1 is expressed at higher levels than SPRR2. Both transcripts are comparable (on a whole skin level) to other published antimicrobial proteins in skin, such as cathelicidin, lysozyme, S100, and the β-defensins. Prior work shows that antimicrobial proteins in the skin and the gut are effective at low μM concentrations (Cash et al., 2006)(Korting et al., 2012), as we show in our current study (Figure 3). Thus, our findings suggest that SPRR proteins are present at similar amounts at the skin surface to other described antimicrobial proteins, which further suggests that the greater infectious burden seen in the *Sprr1a*^−/−^*;Sprr2a*^−/−^mice (Figure 4) is due to the absence of these bactericidal proteins at the skin surface.

**Author response table 1. sa2table1:** 

Gene	Protein	Mean-FPKM	Stdeve
*Sprr1a*	SPPR1	66.30906576	26.04283
*Sprr2a*	SPPR2	6.587556046	3.645252
*Lyz1*	LYSOZYME	36.73892838	32.33246
*Lyz2*	LYSOZYME	255.6400659	234.8976
*Defb1*	DEFENSIN	26.85869015	7.757099
*Defb6*	DEFENSIN	163.2368575	64.78706
*Cramp1*	CATHELICIDIN	7.629683869	0.813505
*S100a8*	S100	2.002705491	0.908623

3. Same question as above can be asked for tissue damage. Of course, these compounds are toxic to human cells, especially in high concentrations at the site of release in the skin. How toxic are these compounds to human cells as compared to the bacteria. How does this relate to the expected concentrations at the site of infections? How does this attribute to inflammation and tissue damage, is it the single cause or is it a small contribution amongst other things? Based on those data, what is the expected therapeutic window (relates to the statement of the authors that these compounds can be used to treat infections).

We thank the reviewer for these thoughtful comments. It has been shown that other antimicrobials in the skin (such as cathelicidin) impact mammalian membranes and stimulate the immune system. We agree that it would be interesting to explore the direct impact of SPRR on mammalian membranes in future studies. Though the biochemistry is not entirely clear, prior work shows SPRR upregulation to be protective for the host (Pradervand et al., 2004)(Vermeij and Backendorf, 2010)(Demetris et al., 2008).

Reviewer #2 (Significance (Required)):4. The antimicrobial action of this group of proteins was already reported, but not in the skin. Authors describe that the depth of penetration is determining the secretion yes or no. This is in line with their findings that all kind of TLR-stimulants do induce expression and secretion, that would include compounds of commensal bacteria. This correlation with depth of penetration means that the way the system discriminates between harmless colonizers and pathogens is by "measuring" the depth of penetration into the skin. This element is novel and of great importance.

We thank the reviewer for their comments and for highlighting the novelty and “great importance” of our study.

References:

Ayabe, T., Satchell, D.P., Wilson, C.L., Parks, W.C., Selsted, M.E., and Ouellette, A.J. (2000). Secretion of microbicidal α-defensins by intestinal Paneth cells in response to bacteria. Nat. Immunol. 2000 12 *1*, 113–118.Cash, H.L., Whitham, C. V., Behrendt, C.L., and Hooper, L. V. (2006). Symbiotic bacteria direct expression of an intestinal bactericidal lectin. Science (80-.). *313*, 1126–1130.Demetris, A.J., Specht, S., Nozaki, I., Lunz, J.G., Stolz, D.B., Murase, N., and Wu, T. (2008). Small proline-rich proteins (SPRR) function as SH3 domain ligands, increase resistance to injury and are associated with epithelial-mesenchymal transition (EMT) in cholangiocytes. J. Hepatol.Harris, T.A., Gattu, S., Propheter, D.C., Kuang, Z., Bel, S., Ruhn, K.A., Chara, A.L., Edwards, M., Zhang, C., Jo, J.-H., et al. (2019). Resistin-like Molecule α Provides Vitamin-A-Dependent Antimicrobial Protection in the Skin. Cell Host Microbe *25*, 777-788.e8.Hu, Z., Zhang, C., Sifuentes-Dominguez, L., Zarek, C.M., Propheter, D.C., Kuang, Z., Wang, Y., Pendse, M., Ruhn, K.A., Hassell, B., et al. (2021). Small proline-rich protein 2A is a gut bactericidal protein deployed during helminth infection. Science *374*.Korting, H.C., Schlölmann, C., Stauss-Grabo, M., and Schäfer-Korting, M. (2012). Antimicrobial Peptides and Skin: A Paradigm of Translational Medicine. Skin Pharmacol. Physiol. *25*, 323–334.Pradervand, S., Yasukawa, H., Muller, O.G., Kjekshus, H., Nakamura, T., St Amand, T.R., Yajima, T., Matsumura, K., Duplain, H., Iwatate, M., et al. (2004). Small proline-rich protein 1A is a gp130 pathway- and stress-inducible cardioprotective protein. EMBO J. *23*, 4517–4525.Vermeij, W.P., and Backendorf, C. (2010). Skin cornification proteins provide global link between ROS detoxification and cell migration during wound healing. PLoS One *5*, e11957–e11957.